# Estimating the burden of underdiagnosis within England: A modelling study of linked primary care data

Olga Anosova[1], Anna Head[1], Brendan Collins[1], Alexandros Alexiou[1],
Kostas Darras[1], Matt Sutton[2], Richard Cookson[3], Laura Anselmi[2],
Martin O'Flaherty[1], Ben Barr[1], Chris Kypridemos[1]*

1 Department of Public Health, Policy and Systems, University of Liverpool, Liverpool, United Kingdom,
2 Health Organisation, Policy and Economics, University of Manchester, Manchester, United Kingdom,
3 Centre for Health Economics, University of York, York, United Kingdom

☯ These authors contributed equally to this work.
* c.kypridemos@liverpool.ac.uk

## Abstract

### Introduction

Undiagnosed chronic disease has serious health consequences, and variation in rates of underdiagnosis between populations can contribute to health inequalities. We aimed to estimate the level of undiagnosed disease of 11 common conditions and its variation across sociodemographic characteristics and regions in England.

### Methods

We used linked primary care, hospital and mortality data on approximately 1.3 million patients registered at a GP practice for more than one year from 01/04/2008–31/03/2020 from Clinical Practice Research Datalink. We created a dynamic state model with six states based on the diagnosis and mortality of 11 conditions: coronary heart disease (CHD), stroke, hypertension, chronic obstructive pulmonary disease, type 2 diabetes, dementia, breast cancer, prostate cancer, lung cancer, colorectal cancer, and depression/anxiety. Undiagnosed disease was conceptualised as those who died with a condition but were not previously diagnosed. This was combined with observed data on the incidence of diagnosis, the case fatality rate in the diagnosed, and an assumption about how that rate varies with diagnosis to estimate the number of undiagnosed disease cases over the total number of disease cases (underdiagnosis) in each population group. We estimated underdiagnosis by year, sex, 10-year age group, relative deprivation, and administrative region. We then applied small-area estimation techniques to derive underdiagnosis estimates for health planning areas (CCGs).

### Results

Levels of underdiagnosis varied between 16% for stroke and 69% for prostate cancer in 2018. For all diseases, the level of underdiagnosis declined over time. Underdiagnosis was

**Data Availability Statement:** The data that we used as an input in our modelling are available from Clinical Practice Research Datalink (CPRD)

and cannot be shared publicly because they contain the electronic health records of pseudonymised patients. CPRD is a real-world research service supporting retrospective and prospective public health and clinical studies. Information on how to get access to the data is available at https://www.cprd.com/data-access. The protocol for this study was approved by the CPRD Independent Scientific Advisory Panel [protocol 20_000096]. Data were from the June 2021 release of CPRD, accessed on 05/10/2021. Access to the surveys we used for the validation is provided through the UK Data Service (https://ukdataservice.ac.uk). The modelling outputs that support the results in our paper are deposited on Zenodo at https://zenodo.org/records/14017677, DOI: 10.5281/zenodo.14017677. We deposited our code at https://github.com/ChristK/Unmet_need.

**Funding:** This project was funded by the National Institute for Health and Care Research (NIHRDH-NIHR130258). https://www.nihr.ac.uk/ PI: BB, Co-Is: CK, LA, MS, RC, MOK The funders had no role in study design, data collection and analysis, decision to publish, or preparation of the manuscript.

**Competing interests:** The authors have declared that no competing interests exist.

not consistently concentrated in areas with high deprivation. For depression/anxiety and stroke, underdiagnosis was estimated to be higher in less deprived CCGs, whilst for CHD and T2DM, it was estimated to be higher in more deprived CCGs, with no apparent relationships for other conditions. We found no uniform spatial patterns of underdiagnosis across all diseases, and the relationship between age, deprivation and the probability of being undiagnosed varied greatly between diseases.

## Discussion

Our findings suggest that underdiagnosis is not consistently concentrated in areas with high deprivation, nor is there a uniform spatial underdiagnosis pattern across diseases. This novel method for estimating the burden of underdiagnosis within England depends on the quality of routinely collected data, but it suggests that more research is needed to understand the key drivers of underdiagnosis.

## Introduction

Underdiagnosis of chronic disease has serious public health consequences and incurs substantial costs to health services. Remaining undiagnosed with conditions and subsequently not receiving early and appropriate treatment might lead to an increased risk of mortality and increased complications [1]. Estimates indicate that relatively large proportions of chronic conditions remain undiagnosed even in countries with relatively accessible health services, such as the United Kingdom (UK). For example, 30% of hypertension and diabetes have been estimated as being undiagnosed in England [2,3]. Understanding how levels of underdiagnosis vary between diseases and health systems is crucial for effectively allocating resources and targeting interventions to increase diagnostic rates. Underdiagnosis has been found to be related to age, gender and socioeconomic status [4,5] and estimating and addressing these differences in access to diagnosis can potentially help reduce inequalities [6–8].

In the NHS, as in many other countries [6], formulae are used to allocate resources to geographical areas with the twin objectives of achieving equal access for equal need and contributing to the reduction of avoidable health inequalities [7]. As these formulae often use information on patterns of diagnosed conditions to estimate relative differences in need between places, differences in access to diagnosis may lead to under-provision of resources to places with high levels of underdiagnosis. This has led to calls for the funding allocation formula to be adjusted to account for these differences in unmet need due to underdiagnosis [7].

Quantifying levels of underdiagnosis is not easy as this is, by definition, unobserved in healthcare records. For a handful of conditions, like type 2 diabetes mellitus (T2DM), hypertension, and mental health, readily available tests exist to diagnose them in the community with reasonable accuracy. Therefore, we can get some estimates of the undiagnosed prevalence of these conditions by surveying the population, although the case definitions might differ between the surveys and clinical practice. Moreover, populations participating in surveys may not be representative of the general population [9]. Such approaches have been used to estimate the proportion of undiagnosed cases of diseases and how this varies across geographical health planning areas in the UK. These have used a combination of national survey data and small-area modelling and then compared this with the prevalence of diagnosed cases either

derived directly from self-reported clinical diagnoses in survey data or with diagnosed prevalence derived from primary care records [10–13].

In parallel, methods have been developed to estimate the burden of undiagnosed prevalence for certain conditions when it is not directly observed. For example, Turakhia and colleagues used a back-calculation methodology to estimate the prevalence of atrial fibrillation (AF), driven by the number of newly diagnosed AF patients immediately after a stroke admission [14]. However, they made the simplifying assumption that diagnosed (likely treated) and undiagnosed (untreated) AF patients have the same annual probability of stroke. On the other hand, Brinks and colleagues came up with a compartment model that describes the disease dynamics but requires specially designed epidemiological studies to inform the parameters of the model [15]. Sporadically, capture-recapture techniques have been used to estimate undiagnosed disease prevalence, although they require strong assumptions to be made [16].

In order to use estimates of underdiagnosis in resource allocation and targeting of interventions within health systems, it is necessary to understand how underdiagnosis varies across subnational health planning areas and across a wide range of diseases and to be able to update these measures regularly to monitor progress. Current approaches are insufficient for this purpose.

We, therefore, developed a new method for deriving estimates of underdiagnosis using linked primary care, hospital and mortality data estimating the burden of undiagnosed disease across 11 common conditions and its variation between local health planning areas in England. The included diseases with a high burden on the UK population according to the Global Burden of Disease project: coronary heart diseases (CHD), stroke, hypertension, chronic obstructive pulmonary disease (COPD), T2DM, dementia, breast cancer, prostate cancer, lung cancer, colorectal cancer, and depression/anxiety.

## Methods

### Data sources and study population

We used the Clinical Practice Research Datalink (CPRD) Aurum database of pseudo-anonymised primary care records from approximately 10% of English GP practices in England registered for more than one year between 1st April 2008 – 31st March 2020. CPRD Aurum is representative of the English population in terms of sex, age, and area-level deprivation [17,18]. The patient data were individually linked to the Hospital Episode Statistics (HES), the UK Office for National Statistics (ONS) Death Registry data, and the Lower Super Output Area (LSOA) dataset of the 2015 English Index of Multiple Deprivation (IMD) based on patients' residential postcodes. The IMD is a composite measure of socioeconomic deprivation at the neighbourhood level [19]. Linked National Cancer Registration and Analysis Service (NCRAS) data on cancer diagnoses were used for validation.

This anonymised linked dataset of approximately 2 million patients was made available to the researcher by CPRD. We excluded individuals from 29 practices because of CPRD warning on duplication of merged practices, individuals who were not eligible for all 3 of HES, ONS, and LSOA linkages, and individuals with the date of censoring being before the date of registration. We only included individuals aged 30 and over. We included only the latest primary care identifier in the study period for individuals with multiple primary care identifiers linked to one HES/ONS identifier. We excluded clinical observation data where dates were recorded outside recorded years of life.

The sample size after cleaning consisted of 1,319,803 individuals, amongst which 1,250,470 had some medical history, and 134,811 had death record information. Patient demographics included sex (male/female), age group (30–49, 50–59, 60–69, 70–79, and 80+ years of age), 9

geographic regions of England (North West, North East, Yorkshire and the Number, West Midlands, East Midlands, East of England, London, South East, South West), and 5 quintile groups of deprivation based on the IMD score of the neighbourhood in which patients lived (1: least deprived; 5: most deprived).

We identified the presence of 11 chronic conditions within the linked CPRD-HES data and as causes of death (primary or contributing) in the ONS mortality data: CHD, stroke, hypertension, COPD, T2DM, dementia, breast cancer, prostate cancer, lung cancer, colorectal cancer, and depression/anxiety. Primary care diagnoses were identified using phenotyping algorithms adapted from a previous project [20]. Secondary care diagnoses (any position in the admission record) and ONS causes of death were identified using ICD10 codes from the HDRUK phenotype algorithms [21]. For the conditions we considered, we assumed that they were lifelong, and the phenotyping algorithms required a concentration of diagnoses to be achieved for depression/anxiety to avoid transient cases being misclassified as lifelong.

### Estimating underdiagnosis

We used a dynamic state model with six states to estimate the number of undiagnosed cases with each disease (Fig 1). From this model, the number of undiagnosed individuals for a specific disease can be estimated from 1) the number of individuals in our data dying of the disease who were not previously diagnosed (undiagnosed deaths), 2) the number of newly diagnosed cases every year, and 3) an assumption about the case fatality rate among the undiagnosed. The first two are directly observed in our data. Essentially, in this model, the people who have died with a condition (as the primary cause of death or contributing factor) but who

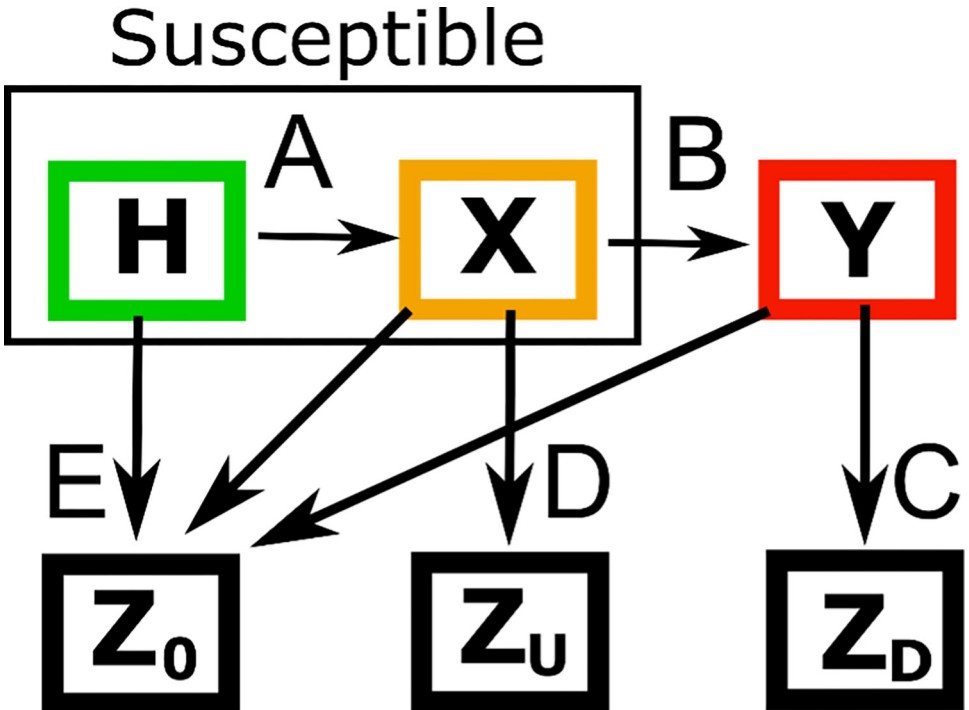

**Fig 1. Model structure.** H denotes the prevalence of healthy people, X is the prevalence of undiagnosed diseased, Y is the prevalence of diagnosed diseased, ZU and ZD are disease-specific deaths for undiagnosed and diagnosed diseased, respectively, and Z0 are deaths from other causes. A, B, C, D, and E are the transition rates between states. Specifically, A denotes the disease incidence rate, B denotes the diagnosis rate, C denotes the case fatality rate among the diagnosed, D denotes the case fatality rate among the undiagnosed, and E denotes the mortality rate from any other cause.

were not previously diagnosed or had only recently been diagnosed are providing an indication of the extent of underdiagnosis in the population. For our main model, we assume the case fatality rate in the undiagnosed is the same as in the diagnosed in the first year of diagnosis, as this is likely to reflect the mortality risk when a disease becomes diagnosable (see S1 Appendix A in S1 Appendix for further justification).

The data provided a sufficient sample to stratify our model by age group, deprivation quintile, sex, region and year, as defined above (i.e. 450 groups per year). Even at this level, however, undiagnosed deaths for many diseases were rare events. For many strata, we observed no undiagnosed deaths simply because of the relatively small size of the strata. We used logistic regression with age, sex, deprivation level and calendar year as predictors to 'impute' the probability of an undiagnosed death for strata with no recorded deaths and to smooth the probability of undiagnosed deaths across the strata. We did not include Region as a predictor to reduce the degrees of freedom of the logistic regressions and enable the models to fit the conditions with a small number of undiagnosed deaths.

We then used dynamic techniques to model transitions between disease states over time (Fig 1). Here, H denotes the prevalence of healthy people, X is the prevalence of undiagnosed diseased, Y is the prevalence of diagnosed diseased, $Z_U$ and $Z_D$ are disease-specific deaths for undiagnosed and diagnosed diseased, respectively, and $Z_0$ are deaths from other causes. A, B, C, D, and E then, are the transition rates between states. Specifically, A denotes the disease incidence rate, B denotes the diagnosis rate, C denotes the case fatality rate among the diagnosed, D denotes the case fatality rate among the undiagnosed, and E denotes the mortality rate from any other cause.

Each transition rate (A, B, C, D, E) varied by individual characteristics (age, sex, deprivation level) and calendar year. All rates varied additionally by Region, except D, as we described above regarding the imputation approach. We assume that each person passes through the undiagnosed state before becoming diagnosed. Changes of state are measured in discrete steps of one-year time intervals.

The undiagnosed prevalence (X) can be estimated from D–the disease-specific case fatality rate in the undiagnosed diseased and $Z_U$—the number of disease-specific deaths amongst the undiagnosed per unit of time. Specifically, $X = \frac{Z_U}{D}$. Furthermore, X is lower-bounded so that enough individuals exist yearly to transition to state Y. We define D as a function of C, separately for each disease, using the case fatality rate among incident-diagnosed cases to define the case fatality rate among the undiagnosed. We first estimated undiagnosed cases and then the probability of being undiagnosed (X/(Y+X)) by sex, 5 age groups, quintile group of IMD (QIMD), English Region and year. For simplicity, from now on, we will use the term underdiagnosis to refer to the probability of being undiagnosed.

## Small area estimates

We used small-area estimation methods to model our regional estimates down to smaller health planning areas, known as Clinical Commissioning Groups (CCGs). CCGs were the main geographical organisations responsible for planning NHS services for their populations across England at this time. We used the 2018 geography for 191 CCGs, which cover the total population of England.

The approach we described above provides estimates of underdiagnosis for a condition by age group, sex, QIMD, and Region. We adopted some of the spatial techniques found within spatial microsimulation and applied an indirect small-area estimation approach, using a geographic model to link probability estimates to a set of predictor variables known for CCG areas [22,23].

To estimate the probability of diagnosis at the CCG level, we compiled auxiliary data at a smaller neighbourhood level, on sex, age group, deprivation level, and the Region that each small area is nested within. These small areas, known as LSOAs, are based on the 2011 Census geography. They have an average population of approximately 1,500 people, and there is a total of 32,844 LSOA areas in England. The compiled table for all LSOAs includes annual population estimates by sex, age group and year between 2008 and 2018, along with the deprivation level of that neighbourhood (QIMD) and the Region within which the LSOA is nested. All data are official statistics provided by ONS and the Ministry of Housing, Communities & Local Government. For every LSOA, we used the probability estimate for each segment and linked them to population estimates for each population group. Probability estimates for each population group within an area were then weighted and aggregated in order to calculate the LSOA-level probability estimates. LSOAs are nested within CCG areas, and as such, we used LSOAs as "building blocks" to further aggregate those and calculate the total CCG probability estimates. In this case, we used 2018 CCG boundaries as a reference geography, but this approach provides the flexibility to summarise results to any administrative boundaries and/or health geography (which are known to change regularly), including CCGs, Integrated Care Systems, or Local Authorities, depending on purpose. We note that similar to small area estimation techniques, the estimation accuracy depends on the number of predictor variables–in this case, since we are using nominal variables, the number of unique population groups. Here, a total of 450 population groups were included for every condition.

## Validation and sensitivity analysis

There is no 'golden' standard that would allow direct comparisons and validation of our estimated underdiagnosis. We, therefore, compared our underdiagnosis estimates for T2DM and Hypertension with estimates from the Health Survey for England and our Depression/Anxiety underdiagnosis estimates with estimates from Understanding Society. We used linked NCRAS data for validation of the four included cancers (breast, colorectal, lung, and prostate). Finally, we used disease-specific emergency admission rates by CCG and broad age groups (30–64, 65+) for the years 2017–2019, and we compared them with our model underdiagnosis estimates of the same years but slightly different age groups (30–59, 60+) due to data limitations. We hypothesised that these two should be positively correlated as CCGs with higher rates of emergency admissions may be less effective in treating known cases and, crucially, identifying new disease cases, leading to underdiagnosis. We calculated the Pearson correlation coefficients between our underdiagnosis estimates for all validations compared to the validation sources.

We additionally conducted sensitivity analyses by applying three alternative modelling assumptions and calculated the correlation coefficient of the distribution of underdiagnosis between CCGs under each assumption relative to our main model assumptions. To test whether using both primary and contributory causes of death, rather than just the primary cause, has a large effect on our estimated distribution of underdiagnosis, we restricted our definition of undiagnosed cases for each disease to those listed as the primary cause of death (as opposed to primary or contributing cause). In our main model, we defined undiagnosed cases as those previously undiagnosed before death. However, expanding our definition to additionally include cases diagnosed within the last year before death could increase the number of undiagnosed deaths. To test whether our estimates are sensitive to using previously undiagnosed cases before death or additionally diagnosed cases within the last year before death, we re-ran the model with the expanded definition of undiagnosed deaths. Finally, we tested the assumption that the case fatality rate in the undiagnosed is the same as the case fatality rate in

**Table 1. Underdiagnosis (probability of being undiagnosed given being diseased), for England 2008 and 2018.**

| Disease | 2008 | 2018 |
|---|---|---|
| Anxiety/Depression | 40% | 29% |
| Breast cancer | 61% | 65% |
| CHD | 52% | 47% |
| Colorectal cancer | 30% | 27% |
| COPD | 40% | 26% |
| Dementia | 41% | 23% |
| Hypertension | 48% | 30% |
| Lung cancer | 48% | 36% |
| Prostate cancer | 52% | 69% |
| Stroke | 22% | 16% |
| Type 2 Diabetes Mellitus | 43% | 24% |

the first year post-diagnosis for the diagnosed. Instead, we assumed that the case fatality rate for the undiagnosed is as that of diagnosed cases overall.

## Results

Table 1 shows the overall levels of underdiagnosis estimated from the model for England as a whole in 2008 and 2018. The proportion of people with disease who were estimated to be undiagnosed ranged from 16% for Stroke and 69% for Prostate Cancer in 2018. Underdiagnosis declined in England during the study period overall and for most diseases, except breast and prostate cancer.

Fig 2 gives the trend of underdiagnosis over time by disease between 2008 and 2018, by level of deprivation. The relationship between deprivation and underdiagnosis varies greatly

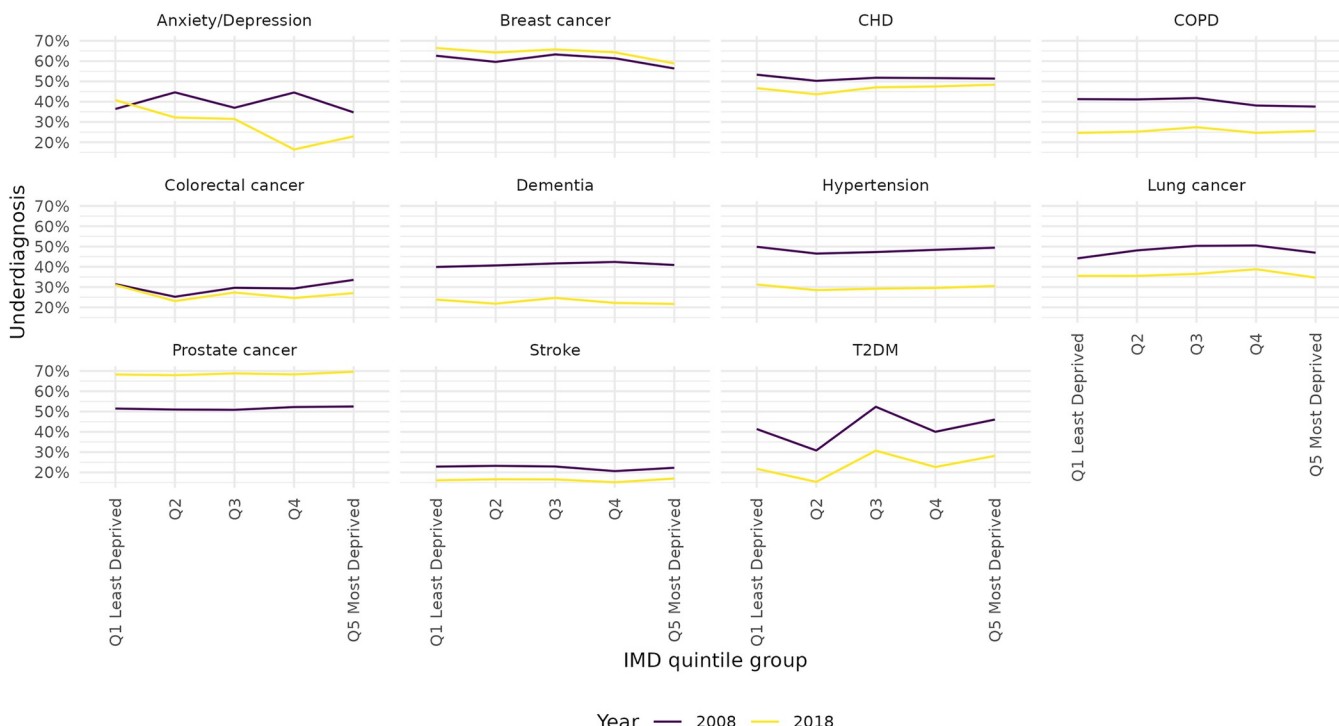

**Fig 2. Underdiagnosis (probability of being undiagnosed given being diseased) by deprivation quintile and disease.** England, 2008 and 2018.

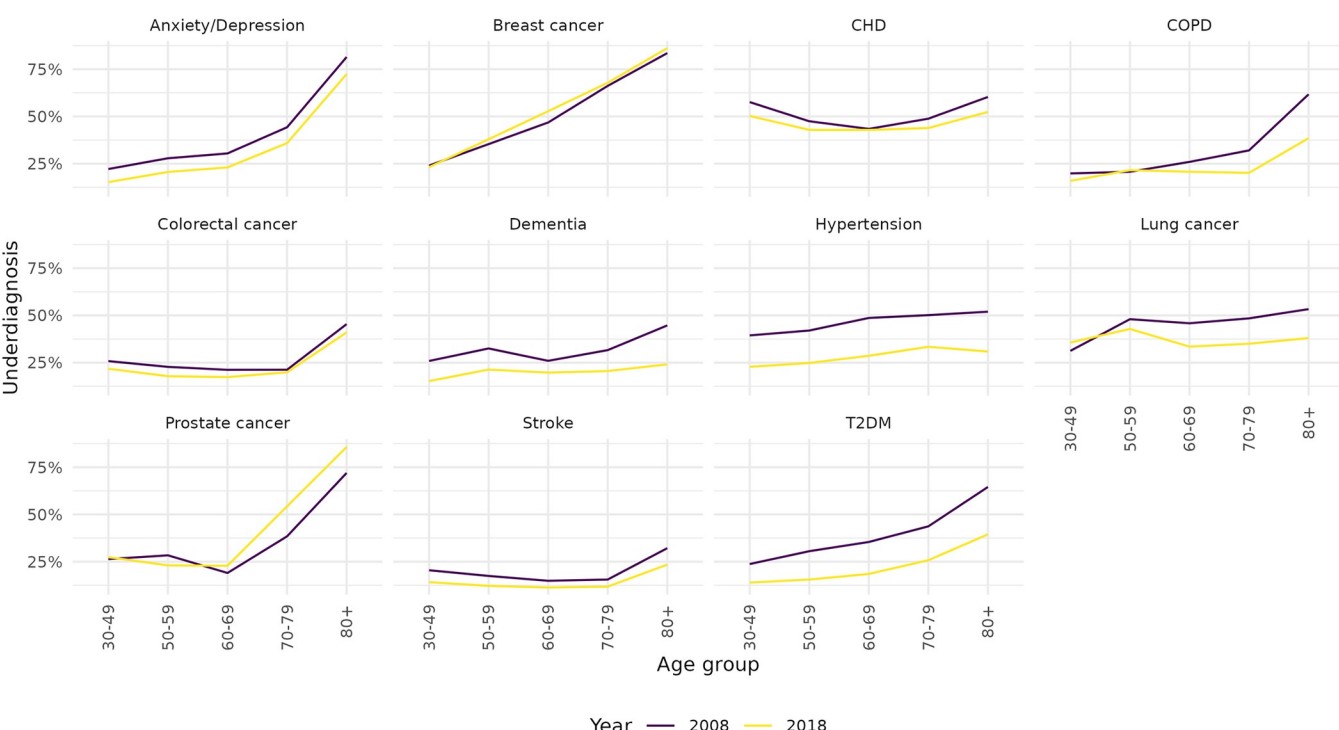

**Fig 3. Underdiagnosis (probability of being undiagnosed given being diseased) by age group and disease in England 2008 and 2018.**

between diseases. For many diseases, there seems to be very little difference in underdiagnosis between levels of deprivation (CHD, Hypertension, Stroke, COPD, Prostate Cancer, Dementia). For Anxiety and Depression, the level of underdiagnosis for the more deprived population (quintiles 4 and 5) declined over time, leading to higher levels of underdiagnosis in less deprived places in 2018. For T2DM, the level of underdiagnosis increases with deprivation.

Fig 3 shows the proportion of undiagnosed by age group in 2008 and 2018. For all diseases apart from CHD, there is a marked increase in underdiagnosis with age. For CHD, the relationship is U-shaped, with underdiagnosis highest at younger and older ages. Estimated levels of underdiagnosis also varied by sex, in particular for Anxiety/Depression, COPD and Hypertension, with men having a higher proportion of underdiagnosis compared to women.

Finally, we show the distribution of the estimated underdiagnosis by CCG (Fig 4). The spatial pattern varies by disease. As an example of the extent of regional differences, some diseases are estimated to be more likely to be undiagnosed in London (COPD and stroke), while others appear to be less likely to be undiagnosed in London (CHD, dementia, hypertension, prostate cancer and T2DM).

## Validation

The probability of being undiagnosed was positively associated with CHD admissions when limited to younger age groups, dementia emergency admissions, prostate cancer, breast cancer (weakly) and diabetes, but negatively associated with emergency admissions for stroke and COPD (see Table 2). For stroke, this could be because many strokes will be first diagnosed through emergency admissions. We find a weak correlation between the probability of being undiagnosed with lung and prostate cancer and the proportion of diagnosis at a late stage.

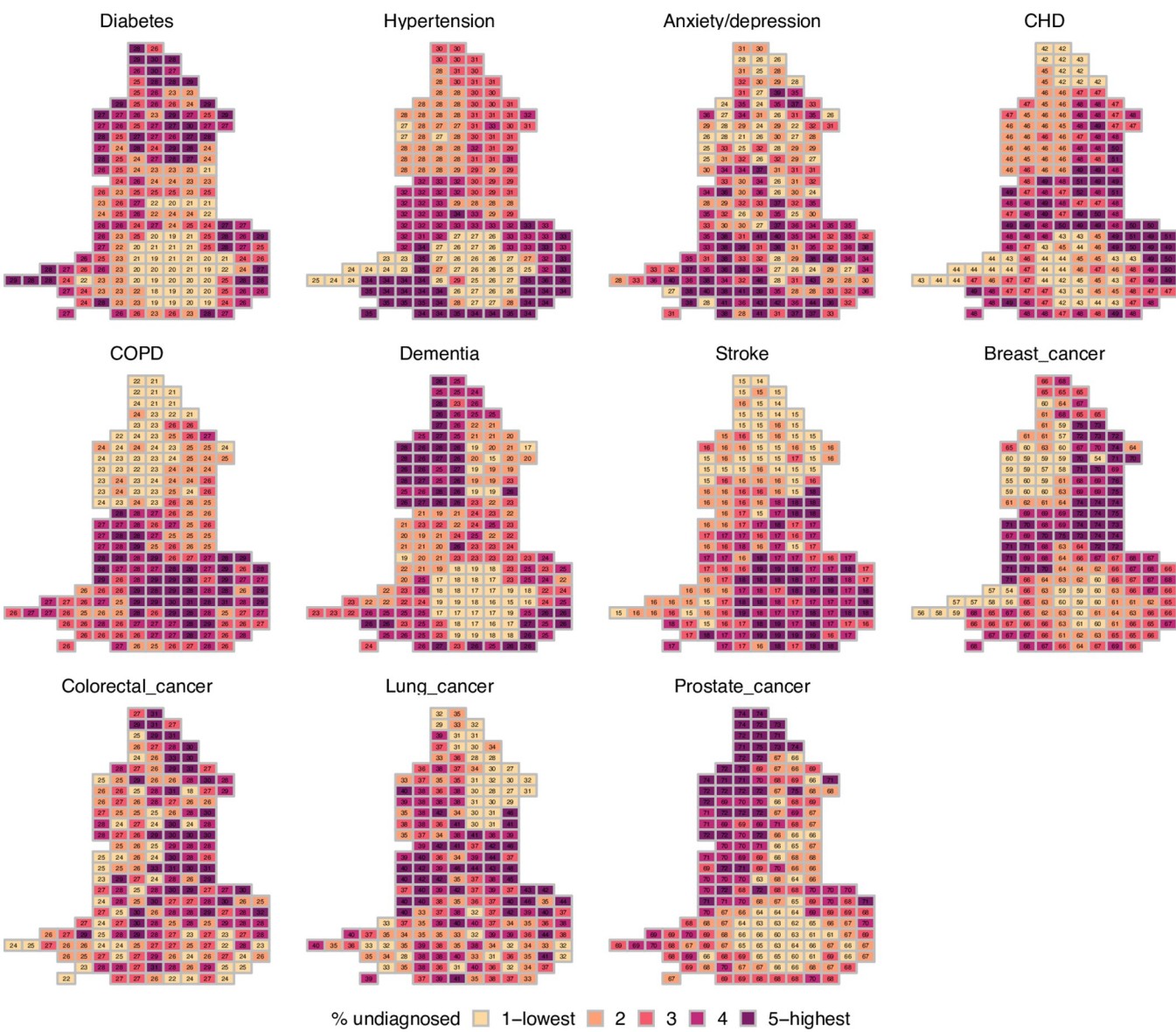

**Fig 4. Estimated proportion of conditions undiagnosed by CCG, 2018 (crude).**

When comparing our estimates from the mortality model to our survey-based estimates (HSE and Understanding Society), we find that for T2DM, there is a strong negative correlation (S3 Appendix C in S3 Appendix). This is because, in our estimates, the probability of being undiagnosed increases with age and deprivation (within age groups), whilst in the HSE, the probability of being undiagnosed decreases with deprivation (within age groups). We see a similar negative association between our depression and anxiety and those derived from Understanding Society (US). This is largely because our estimates indicate a decreasing probability of being undiagnosed with increasing deprivation (within age groups), whilst US indicates that the probability of being undiagnosed increases with deprivation (within age groups). For hypertension, there is no association between our estimates and those derived from HSE. This is partly because our estimates indicate an increasing probability of being undiagnosed

**Table 2. Correlation across CCGs with various comparators.**

| | Corelation between under diagnosis and emergency admissions | | Corelation between under diagnosis and survey based estimates | | Corelation between under diagnosis and late stage diagnosis (stage 4+) | |
|---|---|---|---|---|---|---|
| Disease | rho* | p | rho* | p | rho* | p |
| Anxiety/depression | | | -0.2 | 0.005 | | |
| Breast cancer | 0.16 | 0.025 | | | -0.2 | 0.187 |
| CHD | 0.05 | 0.463 | | | | |
| COPD | -0.59 | <0.001 | | | | |
| Colorectal cancer | 0.06 | 0.449 | | | 0.01 | 0.931 |
| Dementia | 0.29 | <0.001 | | | | |
| Diabetes | 0.38 | <0.001 | -0.55 | <0.001 | | |
| Hypertension | 0.05 | 0.462 | -0.03 | 0.671 | | |
| Lung cancer | -0.16 | 0.024 | | | 0.18 | 0.248 |
| Prostate cancer | 0.45 | <0.001 | | | 0.02 | 0.892 |
| Stroke | -0.18 | 0.014 | | | | |

*rho = Pearson correlation coefficient for underdiagnosis in our main analysis compared to validation sources.

with deprivation, whilst HSE indicates a weak negative relationship with deprivation–i.e. that more deprived people with hypertension are more likely to be diagnosed than less deprived.

## Discussion

In this study, we used linked routine healthcare data to estimate the distribution of being undiagnosed from a group of 11 common conditions. Our results show that 1) underdiagnosis is not consistently concentrated in CCGs with high deprivation, 2) there are no uniform spatial patterns of underdiagnosis across all diseases (i.e. there are no CCGs that systematically underdiagnose all 11 conditions we studied), and 3) underdiagnosis increases with age for most diseases except CHD. While our estimates of underdiagnosis appear higher than estimates from national surveys for T2DM and hypertension but lower than national survey estimates for anxiety and depression, the insights we get regarding patterns of underdiagnosis might still be useful and perhaps more relevant to policymaking than the absolute level of underdiagnosis. These patterns were consistent at large in our sensitivity analysis.

The finding of no consistent pattern with deprivation contradicts other research that has highlighted an inverse care law in access to health care [24,25]. However, in our estimates, underdiagnosis increased substantially with age while life expectancy decreases by deprivation. This may explain the lack of a clear socioeconomic gradient of underdiagnosis. An alternative or perhaps synergistic mechanism could be that people in more deprived areas interact more with the healthcare system, increasing their probability of getting diagnosed earlier. Finally, the inclusion of area deprivation as a predictor in many risk prediction algorithms, i.e. QRISK, QDiabetes, etc., [26,27] may contribute to the inconsistent deprivation patterns.

The second finding of no uniform spatial underdiagnosis pattern across all studied conditions is more difficult to interpret. Leaving aside the possibility that this is an artefact (see limitations below), it may highlight that different CCGs prioritise diagnosing various diseases differently, according to the perceived need in their populations. Unfortunately, this particular research project was not designed to investigate the drivers of these findings, which would require the study of individual CCGs with a mixed methods approach. Future research could use these methods to look at historical priorities (for instance, in regional Sustainability and Transformation Plans or Integrated Care Board Joint Forward Plans) and whether they were associated with a reduction in underdiagnosis of particular conditions.

Previous approaches for estimating the proportion of undiagnosed cases of the disease have estimated the total prevalence of each condition using a combination of national survey data and small-area modelling and then compared this with the prevalence of diagnosed cases derived from primary care records [10–13]. These survey-based approaches have some limitations as they rely on a limited number of variables that are available in national survey data, which are also available for small area populations at sufficient granularity and are updated regularly. They are limited to a few conditions where there is a direct clinical measurement of conditions in representative survey data, generally just T2DM, Hypertension and some mental health conditions. Our approach has some potential advantages over these previous approaches. Firstly, we directly estimate the number of undiagnosed cases of each condition rather than relying on the relative difference between two measures of prevalence (diagnosed and total prevalence) often derived from different data sources using potentially different definitions. Secondly, our estimates are directly estimated from population clinical data rather than assuming that change in disease distribution is solely a function of changes in known risk factors. Thirdly, our approach is not limited to the few conditions for which survey data has clinical measurements. Finally, our estimates can be updated annually using routine electronic health records linked to mortality data.

## Limitations

However, our method has some considerable limitations. Firstly, it may perform better on diseases with high mortality, like cardiovascular disease and cancers, as it is driven by mortality. Unfortunately, we cannot externally validate our estimates for these conditions as we could not identify a suitable source to compare with.

Secondly, it requires large, information-rich datasets, preferably at small locality levels. Although these datasets may exist, access may be problematic due to privacy concerns. For example, the CPRD database contains more than 14m individuals; however, we were only allowed access to a 2m sample. In retrospect, this was inadequate, and we had to impute case fatality rates, especially for younger age groups and for conditions with low mortality, which may have biased our estimates, but avoiding using any produced almost 50% of missing values in the output.

Thirdly, our overall approach is sensitive to the quality and accuracy of death certificates, particularly regarding recording the causes of death. Our analysis is based on the data that precedes the Death Certification Reform program [28], which means that the risk of misclassification is potentially large. The conventional practice of relying on patients' medical records when issuing death certificates may also introduce bias. Differences in referrals to a coroner between different areas could further bias the estimates [28]. Updates to the coding framework used to code cause of death took place in 2011 and 2014, which led to an increase in deaths coded to dementia that would previously be coded as stroke (from 2011) or chest infections (from 2014) [29,30]. To partially overcome the problem of having an uninformative primary cause of death like cardiac or respiratory arrest, especially in older age groups, we considered both primary and contributing causes of death (S3 Appendix C, Table C1 in S3 Appendix).

Fourthly, our approach assumes that medical record-keeping was flawless and there was no misdiagnosis. This may be an unrealistic assumption–for example, there is evidence that the recording of conditions is influenced by clinicians' propensity to code [31] and pay for performance schemes such as the Quality and Outcomes Framework [32] and its impact in terms of direction and magnitude of the bias is hard to estimate.

Finally, the absolute number of undiagnosed cases in our model was highly influenced by the assumption regarding how the case fatality rate in the undiagnosed relates to the case

fatality rate in the diagnosed. However, the distribution of underdiagnosis across the CCGs in England did not change substantially in our sensitivity analysis when we used the case fatality among all diagnosed cases (S3 Appendix C, Table C3 in S3 Appendix). Ideally, this parameter should be informed by empirical evidence, although conducting such epidemiological studies would be perhaps unethical as it would require some participants to remain untreated despite their within-study diagnosis. Practically, this assumption can be informed by expert elicitation and consensus.

## Conclusion

In summary, we presented here an approach to estimate the burden of underdiagnosis using routine linked healthcare data. Our findings suggest that underdiagnosis is not consistently concentrated in CCGs with high deprivation, nor a uniform spatial underdiagnosis pattern across diseases. In the future, we will try to repeat the analysis using a larger dataset, and hopefully, after the implementation of the Death Certification Reform program, that would reduce some of the sources of bias in our research.

## Supporting information

**S1 Appendix. Defining case fatality rate for those undiagnosed.**
(DOCX)

**S2 Appendix. Underdiagnosis by sex and disease, 2008 and 2018.**
(DOCX)

**S3 Appendix. Validation.**
(DOCX)

## Author Contributions

**Conceptualization:** Ben Barr, Chris Kypridemos.

**Data curation:** Olga Anosova, Alexandros Alexiou, Kostas Darras.

**Formal analysis:** Olga Anosova, Alexandros Alexiou, Ben Barr, Chris Kypridemos.

**Funding acquisition:** Matt Sutton, Richard Cookson, Laura Anselmi, Martin O'Flaherty, Ben Barr, Chris Kypridemos.

**Writing – original draft:** Anna Head.

**Writing – review & editing:** Olga Anosova, Anna Head, Brendan Collins, Alexandros Alexiou, Kostas Darras, Matt Sutton, Richard Cookson, Laura Anselmi, Martin O'Flaherty, Ben Barr, Chris Kypridemos.

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
