## [Decision Letter · Decision Letter 0]

29 Jul 2024

PONE-D-24-17631Estimating the burden of underdiagnosis within England: a modelling study of linked primary care dataPLOS ONE

Dear Dr. Kypridemos,

Thank you for submitting your manuscript to PLOS ONE. After careful consideration, we feel that it has merit but does not fully meet PLOS ONE’s publication criteria as it currently stands. Therefore, we invite you to submit a revised version of the manuscript that addresses the points raised during the review process.

We look forward to receiving your revised manuscript.

Kind regards,

Sreeram V. Ramagopalan

Academic Editor

PLOS ONE

Journal Requirements:

4. Please upload a new copy of Figure 4 as the detail is not clear. Please follow the link for more information: " ext-link-type="uri" xlink:type="simple">https://blogs.plos.org/plos/2019/06/looking-good-tips-for-creating-your-plos-figures-graphics/"
" ext-link-type="uri" xlink:type="simple">https://blogs.plos.org/plos/2019/06/looking-good-tips-for-creating-your-plos-figures-graphics/"

Reviewers' comments:

Reviewer's Responses to Questions

**Comments to the Author**

1. Is the manuscript technically sound, and do the data support the conclusions?

Reviewer #1: Yes

2. Has the statistical analysis been performed appropriately and rigorously? 

Reviewer #1: Yes

3. Have the authors made all data underlying the findings in their manuscript fully available?

Reviewer #1: Yes

4. Is the manuscript presented in an intelligible fashion and written in standard English?

Reviewer #1: Yes

5. Review Comments to the Author

Reviewer #1: Dear Editor,

I read the manuscript titled “Estimating the burden of underdiagnosis within England: a modelling study of linked primary care data” with interest. As the authors very eloquently have laid out in the manuscript introduction the study is addressing an important unmet need for healthcare providers and policy makers. They have been very thorough in their modelling. The manuscript is methodologically sound and provides a novel way for exploring and understanding patterns of underdiagnosis in England. This research introduces a novel way of exploring this issue, it highlights the complexity of identifying drivers of underdiagnosis and has highlighted areas where further research is needed.

I only have some very minor comments for the authors.

Methods:

The ‘estimating underdiagnosis section’ could do with some reordering for flow. For example, in line 181 “except D as described above”, I think it should be as described below. The legend to Figure 1 should include definitions for A,B,C,D,E to aid the reader. Potentially consider defining these within the methods text as well.

Discussion

The validation analyses suggest to me that the model overestimates the undiagnosed prevalence for most diseases (except for depression/anxiety). However the aim of the research was not to estimate the prevalence but rather to explore patterns. I think this should be explicit in the discussion. As it mitigates concerns around the magnitude of the problem and concentrates them on identified which populations / areas are most affected.

Table 1. Please clarify that the proportions in the table are out of those that have the disease what proportion is undiagnosed.

Figures:

Quality of Figure 4 needs improving as currently very pixelated.

6. PLOS authors have the option to publish the peer review history of their article (what does this mean?). If published, this will include your full peer review and any attached files.

Reviewer #1: No

---

## [Author Response · Author response to Decision Letter 0]

31 Oct 2024

We have uploaded our response as a separate file

---

## [Editor Report · Decision Letter 1]

4 Nov 2024

Estimating the burden of underdiagnosis within England: a modelling study of linked primary care data

PONE-D-24-17631R1

Dear Dr. Kypridemos,

We’re pleased to inform you that your manuscript has been judged scientifically suitable for publication and will be formally accepted for publication once it meets all outstanding technical requirements.

Kind regards,

Sreeram V. Ramagopalan

Academic Editor

PLOS ONE
---

## [Editor Report · Acceptance letter]

10 Dec 2024

PONE-D-24-17631R1 

PLOS ONE

Dear Dr. Kypridemos, 

I'm pleased to inform you that your manuscript has been deemed suitable for publication in PLOS ONE. Congratulations! Your manuscript is now being handed over to our production team.

Kind regards, 

on behalf of

Dr. Sreeram V. Ramagopalan 

Academic Editor

PLOS ONE